# Structural Analysis of the SANT/Myb Domain of FLASH and YARP Proteins and Their Complex with the C-Terminal Fragment of NPAT by NMR Spectroscopy and Computer Simulations

**DOI:** 10.3390/ijms21155268

**Published:** 2020-07-24

**Authors:** Katarzyna Bucholc, Aleksandra Skrajna, Kinga Adamska, Xiao-Cui Yang, Krzysztof Krajewski, Jarosław Poznański, Michał Dadlez, Zbigniew Domiński, Igor Zhukov

**Affiliations:** 1Institute of Biochemistry and Biophysics, Polish Academy of Sciences, ul. Pawińskiego 5a, 02-106 Warsaw, Poland; kabucholc@gmail.com (K.B.); aleksandra.skrajna@gmail.com (A.S.); kinga.adamska87@gmail.com (K.A.); jarek@ibb.waw.pl (J.P.); michald@ibb.waw.pl (M.D.); 2Department of Biochemistry and Biophysics, University of North Carolina at Chapel Hill, Chapel Hill, NC 27599, USA; kka@med.unc.edu; 3Integrative Program for Biological and Genome Sciences, University of North Carolina at Chapel Hill, Chapel Hill, NC 27599, USA; xcyang@email.unc.edu; 4NanoBioMedical Centre, Adam Mickiewicz University, ul. Wszechnicy Piastowskiej 3, 61-614 Poznań, Poland

**Keywords:** Histone locus body, SANT domain, Myb DNA binding domain, NMR spectroscopy

## Abstract

FLICE-associated huge protein (FLASH), Yin Yang 1-Associated Protein-Related Protein (YARP) and Nuclear Protein, Ataxia-Telangiectasia Locus (NPAT) localize to discrete nuclear structures called histone locus bodies (HLBs) where they control various steps in histone gene expression. Near the C-terminus, FLASH and YARP contain a highly homologous domain that interacts with the C-terminal region of NPAT. Structural aspects of the FLASH–NPAT and YARP–NPAT complexes and their role in histone gene expression remain largely unknown. In this study, we used multidimensional NMR spectroscopy and *in silico* modeling to analyze the C-terminal domain in FLASH and YARP in an unbound form and in a complex with the last 31 amino acids of NPAT. Our results demonstrate that FLASH and YARP domains share the same fold of a triple α-helical bundle that resembles the DNA binding domain of Myb transcriptional factors and the SANT domain found in chromatin-modifying and remodeling complexes. The NPAT peptide contains a single α-helix that makes multiple contacts with α-helices I and III of the FLASH and YARP domains. Surprisingly, in spite of sharing a significant amino acid similarity, each domain likely binds NPAT using a unique network of interactions, yielding two distinct complexes. *In silico* modeling suggests that both complexes are structurally compatible with DNA binding, raising the possibility that they may function in identifying specific sequences within histone gene clusters, hence initiating the assembly of HLBs and regulating histone gene expression during cell cycle progression.

## 1. Introduction

Expression of replication-dependent histone genes is restricted to S-phase of the cell cycle, occurring concomitantly with DNA replication and providing a bulk of histone proteins for *de novo* synthesized DNA [1,2,3,4]. Outside of S-phase, replication-dependent histone genes are transcriptionally repressed and their mRNA products are virtually undetectable in the cytoplasm. Histone mRNAs begin to accumulate to high levels at the G1/S phase transition as a result of enhanced transcription of histone genes and activation of 3’ end processing of the resultant histone pre-mRNAs by U7 snRNP [5].

Transcriptional activity of all classes of replication-dependent histone genes is controlled by Nuclear Protein, Ataxia-Telangiectasia Locus (NPAT). At the G1/S phase transition, this 200 kDa protein is phosphorylated at multiple sites by a complex of cyclin E and cyclin dependent kinase 2 (CDK2), coinciding with the activation of histone gene expression [6,7,8]. The molecular mechanisms that link NPAT and its phosphorylation status with transcription activation of histone genes remain largely unknown [9]. At the G1/S phase transition, the same kinase also phosphorylates the retinoblastoma tumor suppressor protein Rb (pRb) that exists in a dormant complex with the E2F transcriptional factors [10,11]. Hyper-phosphorylation of pRb liberates E2F, which activates transcription of a subset of genes that encode proteins required to support DNA replication [12,13]. Thus, the synchronized phosphorylation of NPAT and pRb by the CDK2/cyclin E complex is important for timely synthesis of the two key chromatin components during the S phase: DNA and histone proteins.

NPAT resides in Histone Locus Bodies (HLBs), nuclear structures that form in the vicinity of histone gene loci [6,7,14,15]. In HLBs, NPAT co-localizes with U7 snRNP, a multi-subunit endonuclease responsible for 3’ end processing of histone pre-mRNAs [16,17,18]. The U7 snRNP consists of an approximately 60-nucleotide U7 snRNA and a heptameric Sm ring characterized by the presence of two unique proteins: Lsm10 and Lsm11 [19,20]. Lsm11 through its extended N-terminal region interacts with the N-terminal region of FLASH, a protein of 220 kDa, and together they recruit a complex of multiple polyadenylation factors to the U7 snRNP [21,22,23]. One of these factors, CPSF73, functions as the processing endonuclease that cleaves histone pre-mRNA at the 3’ end [24]. The resultant mature histone mRNAs are exported to the cytoplasm and used as templates for production of replication-dependent histone proteins.

In human cells, replication-dependent histone genes are clustered on chromosomes 1 and 6 [25]. The cluster on chromosome 6 is significantly larger, containing more than 50 genes encoding all five histone proteins. HLBs that assemble on this chromosome are detected by anti-NPAT antibodies throughout the interphase [6,7], although they are the largest and brightest during S-phase as a result of elevated expression of NPAT and gradual accumulation of other factors involved in histone gene expression, including components of the U7 snRNP [26,27,28]. HLBs that are associated with the smaller histone gene cluster on chromosome 1 can be detected by immunofluorescence with anti-NPAT antibodies only during its peak size in S phase [6,7]. As a result, primary human cells in G1 phase contain only 2 readily detectable NPAT-positive HLBs (those formed on the two copies of chromosome 6) and their number in the S phase increases to 4. Transformed human cell lines due to their aneuploidy contain more HLBs [15,29].

NPAT and subunits of the U7 snRNP are not the only constituents of HLBs. A protein called Yin Yang 1-Associated Protein-Related Protein (YARP) or Gon4l is also enriched in these structures although, in contrast to NPAT and U7 snRNP, it also displays localization throughout the nucleus, indicative of a more general function rather than just regulation of histone gene expression [30,31,32]. How YARP contributes to histone gene expression is largely unknown. Interestingly, YARP associates with a complex of the transcriptional co-repressor Sin3a and histone deacetylase 1 (HDAC1) to represses transcription of specific genes in developing mouse B cells, suggesting that it may work to downregulate expression of histone genes [31]. This possibility is supported by studies on the YARP orthologue in *Drosophila* called *muscle wasted*, which is concentrated in *Drosophila* HLBs. Mutant flies lacking this protein display increased levels of histone mRNAs, consistent with its role as a negative regulator of histone gene expression [28,33].

Near their C-terminus FLASH and YARP contain a highly homologous domain that interacts with the C-terminal region of NPAT. This NPAT-interacting domain folds into a bundle of three α-helices [30] with the overall architecture similar to that of the Myb DNA binding domain and SANT domain found in multiple chromatin-modifying and remodeling complexes [34]. As a first step toward understanding the role of FLASH-NPAT and YARP-NPAT complexes in formation of HLBs and regulation of histone gene expression, we used NMR spectroscopy to determine the 3D structure of the C-terminal fragments of all three proteins in an unbound state. We additionally used a combination of biochemistry and *in silico* modeling to identify potential modes of interactions of NPAT with FLASH and YARP and to analyze the compatibility of these two complexes with DNA binding.

## 2. Results

### 2.1. High-Resolution 3D Structure of FLASH and YARP Fragments

Previous results indicated that the C-terminal fragments of FLASH and YARP fold into three α-helices, structurally resembling the DNA binding domain of the Myb oncogene and the SANT domain found in several chromatin-remodeling and modifying complexes [30]. To characterize these two domains in more detail, we expressed the C-terminal region of FLASH (residues 1923–1982) and the corresponding region of YARP (residues 2149–2209) in bacteria and analyzed their structure by NMR spectroscopy. In FLASH, residue 1982 represents the C-terminal amino acid of the protein. YARP consists of 2240 amino acids and the last 31 amino acids that are not part of the SANT domain were omitted in the analyzed construct.

${}^{1}$H–${}^{15}$N HSQC spectra initially collected for uniformly ${}^{15}$N-labeled domains showed well-dispersed signals for majority amide groups in each domain (Appendix A). The standard procedure based on the analysis of 3D NMR spectra was applied for the assignment of ${}^{1}$H, ${}^{13}$C and ${}^{15}$N backbone resonances in a sequence-specific manner for both fragments [35]. Side-chain ${}^{1}$H and ${}^{13}$C signals were assigned based on 3D HCCH–TOCSY and ${}^{13}$C-edited NOESY–HSQC spectra. The majority of the ${}^{1}$H, ${}^{13}$C, and ${}^{15}$N chemical shifts in aromatic side chains (Tyr, Phe, Trp, His) were achieved by analysis of the ${}^{1}$H-${}^{13}$C HSQC experiments conducted with parameters typical for aromatic groups. As a result, practically all resonances (more than 95%) were assigned in both fragments. The reduced state of the thiol group in cysteines—Cys1941 and Cys1981 in FLASH, and Cys2167 and Cys2207 in YARP—were established based on ${}^{13}$Cβ chemical shifts [36]. The ${}^{13}$Cβ and ${}^{13}$Cγ chemical shifts in proline residues (Pro1946 and Pro1962 in FLASH, and Pro2174 and Pro2189 in YARP) were analyzed with the PROMEGA [37] revealing that all X-Pro peptide bonds were in the *trans* isomeric form.

The secondary structure for the FLASH and YARP fragments was initially predicted by TALOS-N [38] based on backbone chemical shifts and was later confirmed by detecting the characteristic pattern of $d_{\alpha Ni,i+3}$ in the ${}^{15}$N-edited NOESY–HSQC spectra (Appendix A). Structure evaluation procedure used NOESY-derived distance restraints (569 and 644 for FLASH and YARP, respectively), which were additionally supplemented with 136 (FLASH) and 130 (YARP) restraints evaluated for backbone ϕ and ψ torsion angles with TALOS-N (Table 1). Finally, 47 and 40 hydrogen bonds were identified in FLASH and YARP fragments based on geometric criteria [39], leading to 94 and 80 distance restraints introduced in the refined stage of 3D structure solutions. The statistics on the representative ensembles of conformers for both fragments are summarized in Table 1.

The analysis of the 3D structure indicates that each fragment folds into three α-helices, resembling the Myb DNA binding domain and SANT domain [23], consistent with previously published structural reports [30]. In agreement with the significant sequence homology (45% of identical amino acids) (Appendix A), the structures of the FLASH and YARP fragments are very similar with r.m.s.d. of 1.35 Å calculated for Cα atoms of the ordered parts (Appendix A). In FLASH, the location of α-helices are defined as Arg1931–Lys1943 (α-helix I), Phe1948–Lys1957 (α-helix II) and Pro1962–Lys1978 (α-helix III) (Figure 1A). In YARP, the boundaries of α-helices are defined as follows: Arg2157–Glu2169 (α-helix I), Pro2174–Leu2184 (α-helix II) and Pro2189–His2204 (α-helix III) (Figure 1B).

In spite of overall similarity, small differences can be noted between both 3D structures, including a different relative orientation of the α-helices in FLASH and YARP that results from unique interactions within each hydrophobic core. In FLASH the three-helix bundle is defined by the tightly packed aliphatic side-chains of Asp1933, Ile1937, Cys1941, Thr1950, Phe1951, Leu1954, Leu1958, Val1965, and Phe1969 (Appendix A). The corresponding contacts in the hydrophobic core in YARP are formed by Ile2163, Met2166, Cys2167, Thr2176, Phe2177, Ile2180, Ser2181, Leu2184, Val2192, Phe2196, and Leu2199 (Appendix A). The variations in the hydrophobic core contacts lead to alterations in the pattern of inter-helical electrostatic interactions and their contribution to the stabilization of the general fold. In particular, three salt bridges were detected in the FLASH fragment (Asp1433–Lys1460, Asp1434–Arg1468, and Glu1436–Lys1457). The 3D structure of the YARP contains only one salt bridge (Asp2160–Arg2195).

The FLASH and YARP fragments exhibited different distribution of the electrostatic potential (Figure 2). The FLASH fragment is characterized by a highly positive surface formed by Arg1931, Arg1935, and Lys1943 (α-helix I), Arg1944 (loop between α-helices I and II), Lys1949 and Lys1957 (α-helix II), and Lys1960 (loop between α-helices II and III) (Figure 2A). The YARP surface carries more negative potential, with the positively charged regions located in α-helix I (Arg2157, Arg2161) and α-helix II (Arg2195, Arg2197) (Figure 2B).

### 2.2. Backbone Dynamics of the FLASH and YARP Fragments Derived from ${}^{15}$N Relaxation Data

${}^{15}$N relaxation experiments ($R_{1}$, $R_{2}$ and ${}^{1}$H–${}^{15}$N NOE) were performed to compare backbone dynamic processes in the YARP and FLASH fragments. In both domains, six N-terminal and four C-terminal residues display slow transverse relaxation rates ($R_{2}$) accompanied by small or even negative NOE, which typical for disordered regions (Figure 3). The experimental data for the remaining residues, with the ${}^{1}$H-${}^{15}$N NOE values exceeding 0.75 and increased $R_{2}$ relaxation rates, is consistent with the presence of the folded 3D structure. The ${}^{15}$N relaxation data for FLASH resembled that of YARP, indicating that backbone dynamics in both structures are similar.

However, some differences in the dynamic processes between the two proteins should be pointed. In particular, the $R_{2}$ relaxation rates measured for YARP are higher than those obtained for FLASH (Figure 3C,D). Moreover, for YARP, the higher $R_{2}$ values are observed specifically for residues in α-helix I. Moreover, the experimental data measured at 18.8 T demonstrated increase $R_{2}$ for the fragment proximal to Leu2164 and Thr2165 (Figure 3D). A similar but less pronounced effect was observed for residues in α-helix III (Met2200, Gln2201, and Phe2203) (Figure 3D). The detected effect was not observed in FLASH suggesting higher intensity of low-frequency dynamic processes (in ms–μs time-frame) for YARP in compare to FLASH.

Another interesting feature of the backbone molecular dynamic of the two protein fragments is increased flexibility of the loop separating α-helices I and II in FLASH, but not in YARP. Inspection of ${}^{15}$N relaxation data acquired for FLASH indicated low $R_{1}$ and $R_{2}$ relaxation rates and decreased NOE values for residues Ser1947 and Phe1948 (Figure 3A,C,E). Thus, the loop between α-helices I and II exhibited strong dynamic processes in the ns–ps time-frame being much more flexible than any other region of the FLASH. On the other hand, experimental data for the YARP fragment reveal no outliers in the corresponding region (Figure 3B,D,F).

### 2.3. Structure Analysis of the NPAT Peptide in Solution

To study the interactions of the C-terminal fragment of NPAT protein with FLASH and YARP fragments we synthesized a peptide encompassing the last 31 residues of NPAT. The assignment of the NPAT ${}^{1}$H resonances was performed based on the combination of homonuclear 2D TOCSY (mixing time 80 ms) and NOESY (mixing time 120 ms) spectra (Appendix A). ${}^{13}$C and ${}^{15}$N signals were further assigned using heteronuclear 2D ${}^{1}$H-${}^{15}$N and ${}^{1}$H-${}^{13}$C HSQC spectra recorded on natural abundance of the ${}^{13}$C and ${}^{15}$N isotopes (Figure 4A,B).

The high-resolution 3D structure of NPAT peptide was solved using 333 distance restraints yielded from the 2D ${}^{1}$H–${}^{1}$H NOESY spectrum (Table 1). Additional 40 angular restraints for backbone ϕ and ψ torsion angles were obtained with TALOS-N based on ${}^{1}$H, ${}^{13}$C, and ${}^{15}$N chemical shifts. The N-terminal part, comprised of residues ${}^{1397}$KKKKIKKKKL${}^{1406}$, is disordered in solution (Figure 4C). The $d_{\alpha N(i,i+3)}$ contacts detected on the NOESY spectrum (Appendix A) confirmed the existence of an α-helix conformation for the ${}^{1416}$VDKFLLSL${}^{1423}$ fragment folds into a two turn α-helix (Figure 4C,D).

### 2.4. Interactions of the NPAT Peptide with FLASH and YARP Fragments

The residues in FLASH or YARP fragments involved in interactions with NPAT were mapped using chemical shift perturbation (csp) analysis of ${}^{1}$H-${}^{15}$N HSQC spectra upon titration with the NPAT peptide. Step-wise addition of small amounts of the NPAT peptide to protein solutions, resulted in significant changes in the ${}^{1}$H–${}^{15}$N HSQC spectra (See Appendix A). Unfortunately, FLASH–NPAT and YARP–NPAT mixtures are precipitated at equimolar conditions. The best quality NMR spectra were obtained with two fold molar excess of FLASH or YARP, these data were further analyzed.

Upon titration of the FLASH fragment with the NPAT peptide, some FLASH signals dislocated in the ${}^{1}$H–${}^{15}$N HSQC spectrum, while some others broadened or disappeared (Appendix A). The detailed inspection of the experimental data reveals that most of these changes involve residues located in the α-helices I and III (Figure 5A,B). Thus, with the exception of Asp1934, all signals assigned to the residues forming α-helix I (Arg1931-Lys1943) disappear. The same applies to residues located in the C-terminal part of α-helix III (Glu1967-Glu1977), except for Phe1969, Met1973 and Lys1978, which point outward of α-helix I. Residues located in the loop separating α-helices I and II (Gly1945, Ser1947) and in the C-terminal tail succeeding helix III (Ser1979, Lys1980, Cys1981) are also affected by NPAT (Figure 5B). Summarizing, the csp analysis clearly shows that the NPAT-binding site in FLASH includes residues located in α-helix I, the following loop and the C-terminal part of α-helix III that faces α-helix I, as well as the disordered C-terminal tail (Figure 5C).

The same analysis performed for the YARP fragment revealed that most of the changes were observed for residues located in the α-helices I and III, and in the C-terminal tail, mirroring the changes observed in the FLASH fragment (Figure 6). In spite of the general similarity, 5 there are some clear differences between the two NPAT complexes. The NPAT-binding site in the YARP fragment is limited to the central part of α-helix I (Arg2157-Thr2165), while Asp2160 and Ile2163 are unaffected by NPAT binding. Contrary to their counterparts in FLASH, unaffected by NPAT binding were also residues located in the loop between α-helices I and II of YARP (Glu2169–Gly2171) (Figure 6A). The signals assigned to residues located in the central part of YARP α-helix III (Val2192–Arg2197) move in the HSQC spectrum upon NPAT titration rather than disappear, as it is observed in the corresponding fragment of FLASH (Figure 6B). Moreover, nearly entire C-terminal part of the YARP fragment, including Val2192–Gln2201 from α-helix III and the succeeding Leu2202–Glu2208 residues, are involved in binding NPAT (Figure 6C). All these qualitative differences argue that despite overall similarity both complexes differ not only in the exact geometry defined by the relative orientation of the components but also at the level of internal structural dynamics.

### 2.5. Modeling of FLASH-NPAT and YARP-NPAT Complexes

Extreme line broadening of FLASH and YARP signals observed after the addition of NPAT precluded application of any approach based on distance restraints deduced from 3D ${}^{15}$N- or ${}^{13}$C-edited NOESY-HSQC experiments. We were also unable to record ${}^{13}$C-filtered NOESY-HSQC spectra, so the structure of both complexes was modeled *in silico* using qualitative constraints extracted from the csp data. In the modeling procedure, we used a truncated version of the NPAT peptide consisting of 22 residues, lacking the lysine-rich N-terminal region that due to its strong positive charge and flexibility could produce false results. The omitted region is not essential for the interaction with either FLASH or YARP since only the last 17 amino acids of NPAT that encompass the single α-helix and the following acidic region are sufficient for strong complex formation [30].

The NPAT peptide was docked to the FLASH fragment with HADDOCK 2.2 using conditional intermolecular distance constraints between residues located in helices I and III of FLASH and those located in the NPAT helix. The highly scored structures predicted from the modeling clustered with the r.m.s.d. of 1.5 Å and binding energy of −125 ± 11 (kcal mol${}^{-1}$) (Figure 7A). The binding interface of 1708 ± 170 Å${}^{2}$ included the majority of FLASH residues experiencing perturbation in ${}^{1}$H-${}^{15}$N HSQC spectra upon titration with the NPAT peptide. The molecular surface of NPAT and FLASH was reduced as a result of their interaction by a total of 425 ± 25 Å${}^{2}$ when compared to the surface of the two proteins in the unbound state. In the model, the α-helix of NPAT is positioned perpendicularly to the interacting helices I and III of FLASH. In NPAT, the intermolecular interface is formed by all residues of the ${}^{1412}$AGMDVDKFLLSLHY${}^{1425}$ fragment, with the exception of Asp1417, Lys1418, and Leu1421 (underlined in the sequence). In FLASH, the interface is formed by Arg1935, Leu1938, Leu1939 and Gln1942 of helix I, and Phe1969, Leu1972, Met 1973, Leu1975, Phe1976 and Ser1979 of α-helix III. Most of these hydrophobic residues are involved in multiple intermolecular contacts, and all of them reduced FLASH flexibility, as determined by 15 ns trace of MD. According to the model, the two C-terminal residues of NPAT, Asp1426, and Glu1427, might be engaged in a long-range electrostatic interaction with the basic side of FLASH α-helix I (Arg1931, Asn1932 and Arg1935), but the overall contribution of this interaction in formation of a complex between NPAT and FLASH is less significant (Figure 7A).

The model of YARP interacting with NPAT, as predicted by HADDOCK 2.2, generally resembles the NPAT–FLASH model, with some differences existing between the two complexes. The binding of the NPAT α-helix and its flanking amino acids to the SANT/Myb domain of YARP also resulted in a significant reduction of the molecular surface (500 ± 50 Å${}^{2}$). Most importantly, the orientation of NPAT α-helix is almost orthogonal to that observed in the complex of NPAT–FLASH. Despite this difference, the same NPAT residues (Ala1412, Gly1413, Met1414, Val1416, Phe1419, Leu1420, Ser1422, Leu1423, His1424, Tyr1425) are involved in the interaction with the YARP residues Ile2163, Leu2164, Cys2167, Gln2168, Gly2171, Ala2172 of α-helix I and Arg2195, Phe2196, Leu2199, Met2200, Leu2202, Phe2203, Glu2208, Ala2209 of α-helix III, forming a tightly packed hydrophobic interface. As in the NPAT-FLASH complex, the interaction between NPAT and YARP may be stabilized by electrostatic interactions involving the acidic residues located at the C-terminus of NPAT, Asp1426 and Glu1427, and the basic residues from the N-terminal part of YARP α-helix I, Arg2157 and Arg2161.

### 2.6. *In Vitro* Binding

To experimentally validate the *in silico* generated models we introduced a number of rationally selected mutations in the C-terminal region of NPAT that were predicted to contact FLASH and YARP and tested the ability of the resultant NPAT mutants to form complexes with either FLASH or YARP using the GST binding assay. The mutations were made in the context of the last 131 amino acids of NPAT, as previously described [30], with the wild type and mutant constructs being expressed in vitro in the presence of 35S by coupled transcription and translation system (TnT), yielding a protein of ∼15 kDa. FLASH and YARP were expressed in bacteria as fusions with an N-terminal GST tag. In the initial experiments, the FLASH-GST fusion protein (39 kDa) contained the last 103 residues of the full-length FLASH (amino acids 1880–1982) and YARP-GST contained YARP amino acids 2117–2213 [30]. In FLASH–GST, the SANT domain was located close to the C-terminus, whereas in YARP-GST it was followed by additional 29 random amino acids encoded by the vector sequence, providing a convenient way of discriminating this 43 kDa protein from the smaller FLASH fusion.

In the first round of mutagenesis, we replaced leucine at positions 1420, 1421 and 1423 (Leu1420, Leu1421, and Leu1423, respectively) with lysine (Figure 8A) and conducted GST-mediated pull-down experiments in 100 mM KCl. Of these three hydrophobic residues, Leu1420 and Leu1423 were predicted to play a key role in interacting with the hydrophobic core of the SANT/Myb domain in FLASH and YARP (Figure 8B). Consistently, their replacement with lysine residues either individually or together strongly impaired the interaction of NPAT with either protein (Figure 8B,C and Appendix A). The L1420K mutation was more disruptive than the L1423K one, almost completely abolishing the interaction of NPAT with FLASH and YARP. Contrary to L1420K and L1423K mutations, replacing leucine at position 1421 with lysine (L1421K) had no effect on the ability of NPAT to form a complex with FLASH or YARP (Figure 8D), which is consistent with the prediction that this residue does not contact the SANT domain of each protein.

Next, to test whether the acidic C-terminus of NPAT contributes to binding by making electrostatic interactions with positively charged amino acids in the SANT/Myb domain of FLASH and YARP, we substituted Asp1426 and Glu1427 of NPAT with their neutral counterparts, asparagine and glutamine, respectively (Figure 8A). Surprisingly, the double mutation D1426N/E1427Q had no major effect on binding either YARP (Figure 8E) or FLASH. The L1420K/L1423K double mutation tested in parallel as a control abolished the interaction, consistent with the results shown above. We additionally replaced Asp1426 and Glu1427 with positively charged lysine residue or deleted them and no change in binding of NPAT to YARP and FLASH observed (see below), indicating that these two acidic residues have no major role in the formation of each binary complex.

The effects of replacing Leu1420 and Leu1423 were analyzed with another hydrophobic residue, alanine, generating L1420A and L1423A NPAT point mutants (Figure 9A). In this set of experiments, the size of FLASH and YARP in the GST fusions was limited to the SANT/Myb domain, consisting of 60 and 61 amino acids, respectively. Each of the two mutations failed to reduce the ability of NPAT to interact with FLASH (Figure 9B) and almost completely abolished the interaction with YARP (Figure 9C lanes 7 and 11). The discriminatory effect of the L1420A and L1423A mutations on binding of NPAT to FLASH and YARP was observed in three independent experiments (Figure 9D and Appendix A) and is consistent with the models of two complexes relying on at least partially distinct network of interactions and adopting a different conformation, including a different position of the NPAT α-helix relative to the SANT/Myb domain of each protein.

The behavior of the L1420A and L1423A mutations suggests that Leu1420 and Leu1423 play a key role in forming a complex of NPAT with YARP while the relative contribution of these two residues in forming the FLASH–NPAT complex is less important. Initial mapping studies suggested that NPAT interacts with FLASH via both the last 16 amino acids that contain the α-helix and an upstream cluster of lysine residues at positions 1402–1405 [30]. We analyzed the effect of increasing salt concentration on formation of the two NPAT complexes and observed that while the NPAT–YARP complex was stabilized at 600 mM KCl, the interaction of NPAT with FLASH was weakened at this salt concentration (Appendix A). This further supports the notion that the two complexes utilize partially distinct network of interactions and argues that there is a significant contribution of electrostatic contacts in formation of the NPAT–FLASH complex, likely involving lysine 1402–1405, which may mask the negative effect of the L1420A and L1423A mutations. Importantly, either mutation was sufficient to disrupt the NPAT–FLASH complex when the GST binding assay was carried out at 200 mM KCl (Appendix A). Thus, Leu1420 and Leu1423 are important for the formation of a hydrophobic interface in both complexes, although the relative contribution of these residues is less important in formation of the NPAT–FLASH complex and readily detectable only in the presence of high salt concentration that eliminates accompanying electrostatic interactions.

In order to find additional discriminatory mutations in NPAT, we individually replaced Phe1419 and Tyr1425 with alanine (F1419A and Y1425A, respectively), both involved in interacting with the hydrophobic cores formed by α-helices I and III of FLASH and YARP. Of these two mutations, F1419A had a similar inhibitory effect on the formation of either complex, whereas Y1425A predominantly affected the interaction of NPAT with YARP and slightly enhanced the interaction with FLASH (Figure 9C(lanes 5–8 and 17–20),D and Appendix A), thus mirroring the effect of L1423A. As indicated below, deleting the two acidic residues at the C-terminus of NPAT (ΔDE) produced an NPAT variant that retained its normal interaction with both YARP and FLASH (Figure 9C(lanes 21–24),D and Appendix A). Other mutations in NPAT, including F1410A, V1419T, H1424A, behaved in the same manner as F1419A, equally inhibiting formation of both complexes.

The effects of point mutations on interactions between proteins in FLASH–NPAT and YARP–NPAT complexes were additionally analyzed by calculations of changes of free energy (ΔΔG) (Appendix A). The differences obtained with computer modeling are in line with mutation experiments and modeled 3D structures of FLASH–NPAT and YARP–NPAT complexes. For YARP–NPAT interactions, the L1423A, and F1419A mutants reveal a larger change in ΔΔG. Performed calculations select the F1419A mutation as the most valuable for the FLASH–NPAT complex (Appendix A). At the same time, we note that L1423A mutation does not fit the exponential relation suggesting that Leu1423 probably does not interact with the FLASH fragment.

### 2.7. Can FLASH and YARP Bind DNA?

The major distinction between the Myb and SANT domains is that the latter lacks a positively charged surface in the third recognition helix and often contains amino acids that are incompatible with DNA binding [34]. We tested the ability of FLASH and YARP to interact with DNA by molecular docking, superimposing our NMR structures of FLASH and YARP alone or in a complex with NPAT onto the available structures of the Myb DBD/DNA complex (see Materials and Methods). In the top scoring model of FLASH bound to DNA, Lys1949 from helix II and Arg1968, Lys1974 and Lys1978 from α-helix III are involved in a strong electrostatic interaction with the DNA backbone, additionally supplemented by an H-bond formed by Gln1971. Interestingly, a tract of H-bond donor(s)/acceptor(s) formed by Asn1963, Glu1967 and Gln1970, all of which are located inside the DNA major grove, may mediate a sequence-specific interaction with DNA. However, no particular sequence preferences can be deduced from this model. Additionally, this FLASH–DNA complex is stabilized by numerous hydrophobic contacts involving Phe1948, which are located in the minor grove.

In the YARP–DNA complex, only two basic residues from α-helix III (Arg2195 and Arg2197) form salt bridges with the DNA backbone. The interaction is additionally supported by H-bonds formed by Gln2173 (loop 1), Asn2178 (helix II) and Gln2201 (helix III), but slightly disfavored by Glu2198 (helix III) located proximal to the DNA phosphate groups. This residue together with His2194 and Arg2197 from helix III is placed deeply in the major grove, potentially playing a role in sequence-specific DNA recognition by YARP. Based on these models we conclude that both FLASH and YARP may bind directly to DNA in a sequence-specific Myb-like manner, with the pattern of protein-DNA contacts identified *in silico* favoring FLASH over YARP as a DNA-binding protein.

The position of NPAT in the complex with FLASH and YARP indicates that it is unlikely to interfere with binding of either protein to DNA. On the contrary, the NPAT peptide, in spite of interacting in a different orientation in both complexes, may facilitate binding of FLASH and YARP to DNA by contributing the extremely basic N-terminal ${}^{1397}$KKKKIKKKK${}^{1406}$ region for multiple contacts with the DNA-phosphate backbone. This sequence is not included in the model but as a result of its flexibility it is likely to be properly positioned in both complexes to at least partially interact with DNA. Interestingly, a lysine-rich sequence at this position is a common feature of all known NPAT orthologues in vertebrates, being an indicative of important and conserved function.

Structures of FLASH and YARP in complex with DNA modeled by homology to the Myb/DNA complexes only slightly differ from those obtained from molecular dynamics based on the NMR-derived structures of FLASH and YARP (Appendix A). This clearly confirms that the fold of both protein domains determined in solution is compatible with their interaction with DNA.

## 3. Discussion

FLASH and YARP, two proteins with the molecular mass of more than 200 kDa each, are key regulators of histone gene expression in animal cells [30,46]. FLASH plays a well-documented role in 3’ end cleavage of histone pre-mRNAs, an essential and highly specialized processing event that occurs co-transcriptionally in the nucleus and generates mature histone mRNAs [21]. The role of YARP is less understood but studies on mouse and *Drosophila* YARP orthologues suggest that it may act as a negative factor to repress histone gene transcription [31,33]. Both FLASH and YARP contain near the C-terminus a highly homologous domain of approximately 60 amino acids that is capable of interacting with the C-terminal region of NPAT [30], a protein of 150 kDa and a key transcriptional activator in histone gene expression [6,7,15,29]. To get initial insight into the role of these interactions in histone gene expression, we determined the solution structures of the C-terminal regions of FLASH, YARP and NPAT by NMR spectroscopy and identified potential modes of interactions in the FLASH–NPAT and YARP–NPAT complexes by *in silico* modeling.

### 3.1. FLASH, YARP and Their Complexes with NPAT

Our NMR studies demonstrate that the C-terminal domain of FLASH and YARP adopts the same fold that consists of three α-helices and is highly similar to the DNA binding domain (DBD) of the Myb transcriptional activator and the SANT domain found in various subunits of larger complexes involved in chromatin remodeling and histone modification [34]. As in the Myb and SANT domains, the overall structural fold of the FLASH and YARP domain is maintained by a hydrophobic core contributed by residues from all three α-helices. The C-terminal region of NPAT contains a single and relatively short α-helix that uses a subset of hydrophobic residues to tightly pack against α-helices I and III of FLASH and YARP. According to our NMR results and *in silico* modeling, FLASH and YARP in spite of sharing the same fold and ∼40% identical amino acids (human proteins) bind NPAT using partially distinct network of interactions. As a result, NPAT is placed in the two complexes in a different orientation relative to FLASH and YARP, a surprising conclusion given the structural similarity and high degree of amino acids conservation between the SANT/Myb domains of FLASH and YARP.

Mutating Leu1420 and Leu1423 to lysines either alone or in combination strongly compromised the ability of NPAT to interact with FLASH and YARP, pointing to the importance of hydrophobic residues in these two positions of the NPAT α-helix in contacting the SANT domain. In contrast, the same substitution of Leu1421 had no effect. In the NMR-based structure, this lysine does not contact residues of α-helices I and III of the FLASH/YARP domain, facing instead a different direction. Consistently, this residue is less conserved than the remaining amino acids of the NPAT helix. Interestingly, the substitution of the essential lysines 1420 and 1423 with alanines selectively reduced binding of NPAT to YARP having at the same time no major effect on binding to FLASH. The same discriminatory behavior was observed for the Y1425A mutation. Together, the effect of these NPAT mutations is consistent with our conclusions based on the *in silico* modeling that NPAT binds helices I and III of FLASH and YARP using at least partially different network of contacts, resulting in each complex adopting a unique conformation likely determining its specific functions in histone gene expression. Further structural and mutagenesis studies, including X-ray crystallography, are required to identify which amino acids in FLASH and YARP are responsible for the unique orientation of the NPAT α-helix in each complex.

Various mutations of the two acidic residues at the C-terminus of NPAT, Asp1426 and Glu1427, including their deletion or substitution with positively charged amino acids, had no effect, suggesting that they do not play any major role in binding FLASH and YARP. These two residues are ubiquitously conserved in all known vertebrate NPAT orthologues, and they may play an important role in a different interaction.

### 3.2. Myb and SANT Domains Are Multi-Functional Binding Modules

The Myb transcriptional activator is a sequence-specific DNA binding protein and a known proto-oncogene that recognizes the AACNG motif found in the promoters of Myb-controlled genes [47,48]. It contains three tandemly arranged imperfect repeats of the Myb DNA binding domain (DBD R1-3), each consisting of three α-helices [48]. The sequence-specific recognition of the AACNG motif is mediated by the C-terminal α-helix which involves base-specific interactions within the major grove and multiple contacts with the DNA-phosphate backbone. The SANT domain was initially defined as a domain of unknown function that shares structural similarities with the Myb DBD and exists in components of multiple chromatin remodeling and histone modifying complexes involved in transcriptional regulation [49]. The third (recognition) helix in many SANT domains contains negatively charged amino acids that are incompatible with the canonical mode of DNA binding displayed by the Myb domain and the SANT domain of some proteins was shown to interact with positively charged histones rather than DNA [50,51]. Based on these observations, it was hypothesized that the SANT domain may function as a specialized histone-tail recognition platform [34].

A growing body of evidence indicates that the family of the Myb/SANT domains sharing the same triple α-helical bundle is functionally more diverse and capable of interacting with multiple partners. For example, the two helices of R2 of the Myb transcriptional activator that do not contribute to DNA binding interact instead with the N-terminal tail of histones H3 and H3.3, promoting their acetylation [52]. The role of the Myb R1, which also does not participate in DNA recognition, is unknown but its deletion enhances the oncogenic potential of v-Myb, suggesting that it may bind another protein [53,54]. A different example of the functional versatility of the SANT/Myb fold is provided by two nuclear receptor co-repressors, SMRT (silencing mediator of retinoid acid and thyroid hormone receptor) and NCoR (nuclear receptor co-repressor). Each of these proteins contains two SANT domains, with one copy catalytically activating histone deacetylase 3 (HDAC3) [55,56] and the other copy interacting with core histone tails [57]. The same structural and functional arrangement was identified in the Co-REST co-repressor that interacts with histone deacetylases 1 and 2 (HDAC1-2) [57,58]. Some other proteins contain only a single SANT domain that appears to have a dual function, binding both histone substrate and acetlytransferase, resulting in its catalytic activation [50,51].

These and other observations indicate that the SANT/Myb domain represents a versatile structural unit that employs all three helices for multiple interactions that extend beyond binding double stranded DNA and histone proteins, often co-operating with other structural elements in the same protein to assemble multi-component complexes. Our work adds the C-terminal region of NPAT to the list of binding partners of the SANT/Myb domain and demonstrates that by binding helices I and III of FLASH it likely contributes to forming a unique platform to regulate expression of replication-dependent histone genes during the cell cycle.

### 3.3. FLASH-NPAT Complex as a Platform That Recognizes Histone Gene Clusters

The mechanisms by which the FLASH–NPAT and YARP–NPAT complexes participate in regulation of histone gene expression remain unknown. All three proteins are highly enriched in Histone Locus Bodies [29,30], macroscopically detectable structures that form in the nucleus in the vicinity of histone genes [59]. One critical event in the biogenesis of the HLBs that remains poorly understood is how clusters of replication-dependent histone genes are recognized in the genome, giving rise to localized enrichment of multiple components of the specific transcriptional and processing machinery that generates mature histone mRNAs [28].

Studies in mammalian cells demonstrated that the interaction between the C-terminal regions of FLASH and NPAT is essential for formation of HLBs and that down-regulation of either protein by RNAi results in rapid mislocalization of the other protein to the nucleoplasm and its proteolytic degradation [29,30,60]. Thus, FLASH and NPAT are unstable as individual proteins and likely act in conjunction to assemble/maintain HLBs. Downregulation of FLASH and NPAT in *Drosophila* also prevents the assembly of HLBs [28,33]. The effect of depletion of YARP on HLB formation in mammalian cells has not been investigated. However studies in flies and *Drosophila* cultured cells demonstrated that in the absence of the *Drosophila* orthologue of YARP, muscle wasted, the HLBs are detected normally, suggesting that YARP in contrast to FLASH is not essential for the HLB assembly. Thus, the interacting FLASH and NPAT may form together a unique platform located near the C-terminus of both proteins that serves to recognize histone gene clusters, initiating the assembly of HLBs.

The DNA binding domain of the Myb transcriptional activator in spite of using two cooperatively acting α-helices can specifically recognize only relatively short and partially degenerate sites in DNA [48,61]. Other Myb-related proteins that contain only one Myb DBD, including human TRF1, TRF2 [62] and *Drosophila* Adf-1 [63], enhance their site-specific DNA binding potential by protein dimerization allowing for simultaneous recognition of two tandemly arranged target sequences [62]. While our modeling analysis indicates that the SANT/Myb domain of FLASH, both alone and in a complex with NPAT, may be capable of binding DNA, it likely requires additional mechanisms to specifically identify histone gene loci and prevent mislocalization of HLB components to illegitimate sites in the genome. One possibility is that the ability of FLASH–NPAT complex to site-specifically recognize histone gene loci may be facilitated by oligomerization of full length NPAT [64] and co-operative binding of multiple recognition units to yet uncharacterized nucleotide sequences in clusters of replication-dependent histone genes. However, we favor an alternative hypothesis in which the increased level of specificity is achieved by the FLASH–NPAT complex additionally recognizing unique modifications or other structural features of chromatin that might exist in the vicinity of histone genes. This possibility is consistent with the known function of some SANT domains as histone recognition modules [34] and our NMR-based structure demonstrating that various regions of the complex, including the internal helix of FLASH, are potentially available for other interactions. As previously proposed in a hypothetical model [46], following the recognition of histone gene loci by the complex of FLASH and NPAT and formation of HLBs during G1 phase, NPAT is hyper-phosphorylated during the onset of S phase by cyclin E/Cdk2 complex [6,7], resulting in separation of the complex and simultaneous activation of histone gene transcription and 3’ end processing of histone pre-mRNAs by the liberated NPAT and FLASH (as part of U7 snRNP), respectively [46].

More difficult to explain is the role of YARP in histone gene expression and whether YARP competes with FLASH for binding NPAT or the two complexes are formed during different phases of the cell cycle. The SANT/Myb domain in YARP, while sharing with the corresponding domain in FLASH ∼40% of identical amino acids, the same structural fold and the ability to interact with the C-terminal peptide of NPAT, also displays a number of unique features. Analysis of the electrostatic potential surfaces in FLASH and YARP domains revealed different charge distribution in the two proteins, with YARP being overall more acidic. Another feature that distinguishes the two domains is the presence of a single glycine in the turn between α-helices II and III in FLASH but not in YARP [30]. Finally, the SANT/Myb domain in FLASH is located at the extreme C-terminus, whereas in YARP of all vertebrates and *sea urchins* it is followed by additional ∼30 amino acids. This C-terminal tail of YARP is predominantly acidic, adding a strong negative charge to the vicinity of the domain.

The unique features of the SANT/Myb domain of YARP and the different mode of its interaction with NPAT likely determine binding partners of the YARP–NPAT complex, hence contributing to its functional specialization in histone gene expression. Studies on the *Drosophila* orthologue of YARP, muscle wasted, suggest that it may act as a transcriptional repressor preventing over-expression of histone genes during development and cell cycle [33]. Thus one possibility is that YARP interacts with NPAT at the end of S phase, with the two proteins binding active histone genes and down regulating their expression in G2 phase. A region in YARP was shown to interact with the Sin3a deacetylation complex [31], providing a plausible mechanism for the repression.

## 4. Materials and Methods

### 4.1. Expression and Purification of the C-Terminal Fragments of FLASH and YARP for NMR Studies

C-terminal fragments of FLASH (residues 1923–1982) and YARP (2149–2209) were expressed in the *E. coli* BL21 (DE3) strain from the pET28a vector using T7 expression system on minimal M9 growth media. The two proteins were uniformly labeled with either ${}^{15}$N or ${}^{13}$C${}^{15}$N using ${}^{15}$NH${}_{4}$Cl and U-${}^{13}$C${}_{6}$ glucose (both EurisoTop, Paris, France) as a source for nitrogen and carbon isotopes and purified via the N-terminal 6xHis tag on nickel beads (Qiagen Pharma, New-York, NY, USA), as recommended by the manufacturer. The FLASH and YARP fragments were further purified by Size Exclusion Chromatography using a Superdex 75 column (GE Healthcare, Warsaw, Poland) concentrated, and supplemented with 40 mM TCEP and 50 mM Arg+Glu.

### 4.2. Synthesis and Purification of the C-terminal NPAT Peptide

The synthetic peptide KKKKIKKKKLPSSFPAGMDVDKFLLSLHYDE (free N-terminus carboxyl at C-terminus) corresponding to residues 1397-1427 of NPAT (Uniprot Q14207) was synthesized at High-Throughput Peptide Synthesis and Array Core Facility at UNC using Fmoc solid phase peptide synthesis. The peptide was synthesized at 25 μM synthesis scale on PTI Symphony automated peptide synthesizer using H-Glu(tBu)-HMPB ChemMatrix resin (substitution 0.49 mM/g). Aspartic acid residues were introduced using Fmoc-Asp(OMpe)-OH. The peptide was cleaved from the resin and deprotected using 2.5% water, 2.5% triisopropylsilane in trifluoroacetic acid, precipitated and washed with cold diethyl ether, air dried and lyophilized from 50% acetonitrile. After lyophilization the crude peptide was purified on preparative RP-HPLC (Waters SymmetryShield RP-18 column) and lyophilized. The identity and quality of the resulting peptide was confirmed by analytical HPLC and MALDI-TOF MS.

### 4.3. Multidimensional NMR Spectroscopy

All NMR samples were prepared in 25 mM TRIS-HCl buffer (H${}_{2}$O:D${}_{2}$O 9:1 v/v) in the presence of 200 mM KCl for FLASH and 250 mM KCl for YARP samples at pH 6.2. 50 mM Arg+Glu and 40 mM Tris(2-carboxyethyl)phosphine (TCEP) were additionally added to FLASH and YARP solutions to improve protein solubility and prevent cysteine oxidation.

All NMR experiments were carried out on Agilent DDR2 800 NMR spectrometer operated at 18.8 T (${}^{1}$H resonance frequency 799.94 MHz) equipped with four channels, Performa IV z-gradient unit and inverse ${}^{1}$H/${}^{13}$C/${}^{15}$N probe. The NMR spectra were acquired at 298 K, processed with NMRPipe software [65] and further analyzed in the Sparky program [66]. The ${}^{13}$C and ${}^{15}$N chemical shifts were referred in the indirect manner according to the external sodium 2,2-dimethyl-2-silapentane-5-sulfonate (DSS) reference using Ξ = 0.251449530 and 0.101329118 ratios for ${}^{13}$C and ${}^{15}$N nuclei, respectively [67].

In case of the FLASH and YARP fragments ${}^{1}$H, ${}^{13}$C, ${}^{13}$Cα, ${}^{13}$Cβ, and ${}^{15}$N resonances were assigned for uniformly ${}^{13}$C,${}^{15}$N-labeled samples using the standard combination of heteronuclear 3D spectra (HNCO, HN(CA)CO, HN(CO)CA, HNCA, CBCA(CO)NH and HNCACB) [35]. The ${}^{1}$H and ${}^{13}$C side-chain resonances were obtained from analysis of the 3D C(CO)NH, H(CCO)NH and HCCH-TOCSY NMR data.

The assignments of the ${}^{1}$H, ${}^{13}$C, and ${}^{15}$N resonances in NPAT peptide were obtained on the basis of 2D homonuclear (TOCSY, NOESY) NMR spectra recorded with various mixing times. These data were supplemented with 2D heteronuclear (${}^{1}$H-${}^{15}$N HSQC, ${}^{1}$H-${}^{13}$C HSQC) experiments acquired on natural abundance of the ${}^{13}$C and ${}^{15}$N isotopes.

### 4.4. Evaluation of High-Resolution 3D Structures of the FLASH, YARP, and NPAT Fragments

In case of FLASH and YARP fragments, 3D ${}^{15}$N- and ${}^{13}$C-edited NOESY-HSQC spectra were acquired for ${}^{15}$N- or ${}^{13}$C,${}^{15}$N-labeled forms. As a result, 541 (219 intra-residue, 147 sequential, 94 medium range, and 81 long range) and 606 (224 intra-residue, 201 sequential, 141 medium range, and 40 long range) distance constraints were yielded for FLASH and YARP, respectively (Table 1). These data were additionally supplemented with angular restraints deduced from the chemical shifts of ${}^{15}$N, ${}^{13}$C, ${}^{1}$H, ${}^{13}$Cα and ${}^{13}$Cβ resonances using TALOS-N software [38]. In the final rounds of simulations, each identified hydrogen bond was converted to restraints for r${}_{NH-O}$ (1.5 – 2.8 Å) and r${}_{N-O}$ (2.4 – 3.5 Å) distances [39]. The initial step of calculations was performed using the CYANA (version 3.95) software [68]. The refinement with an explicit solvent was conducted for the 20 highest-scored structures with the aid of YASARA software [42] using Yasara2 force-field and isothermal–isobaric (NPT) conditions. The resulting set of 20 structures was finally validated with PROCHECK-NMR [40] and MolProbity [69] programs, and visualized with either MOLMOL [70], Chimera [71], or YASARA software.

### 4.5. ${}^{15}$N Relaxation of the Backbone Amide Groups

Longitudinal ($R_{1}$) and transverse ($R_{2}$) ${}^{15}$N relaxation rates, together with ${}^{1}$H-${}^{15}$N heteronuclear NOE enhancement were determined for uniformly ${}^{15}$N-labeled FLASH and YARP samples on Varian Unity+ 500 and Agilent DDR2 800 spectrometers at 11.7 and 18.8 T, respectively. The standard pulse sequences [72] included in BioPack software (Agilent Inc., USA) were applied. The longitudinal magnetization was recorded at ten evolution times: 10, 90, 170, 290, 410, 550, 690, 850, 1010 and 1250 ms. Transverse relaxation was measured with Carr-Purcell-Meiboom-Gill (CPMG) pulse train (refocusing time of 650 μs) at 10, 30 50, 70, 90, 130, 170, and 210 ms evolution times. The cross-correlation effect was suppressed by the delay between π(${}^{1}$H) pulses of 5 and 10 ms for $R_{1}$ and $R_{2}$ measurements, respectively [73]. The steady-state ${}^{1}$H–${}^{15}$N NOE was measured with 6 s delay. All relaxation rates were estimated using a two-parameter model of a single exponent decay with the RELAX software [74] (version 4.0.1). Performed NMR experiments provide to extract $R_{1}$, $R_{2}$ and ${}^{1}$H-${}^{15}$N NOEs for 45 and 54 residues of FLASH and for 52 and 55 residues of YARP at 11.7 T and 18.8 T, respectively.

### 4.6. Solution 3D Structures of the FLASH-NPAT and YARP-NPAT Complexes

Both structures were modeled *in silico* using the HADDOCK 2.2 software [75] available at the European WeNMR grid facility (milou.science.uu.nl/services/HADDOCK2.2). The 3D structures of YARP, FLASH, and NPAT fragments were used as templates for the modeled complexes. Weak conditional distance restraints between C-terminal residues of NPAT and FLASH/YARP residues, located in α-helices I and III and affected upon NPAT binding, were introduced to drive the relative orientation of components in the complex. After the standard assessment, the resulting structures were additionally evaluated according to the agreement with the contact surface deduced from the csp data. For both complexes, the best models were then subjected to simulated annealing procedure performed in the presence of explicit water with the aid of YASARA software. The initial 1 ns of simulations were performed at 500 K with all hydrogen bonds within helical regions constrained by the appropriate distance restraints introduced to prevent uncontrolled unfolding. During the next 1 ns, the system was cooled down to 100 K with the weights for the distance restraints being gradually decreased to zero. Five repetitions of the above procedure were followed by 40 ns of unconstrained molecular dynamics at 300 K.

The structural stability of YARP and FLASH and their complexes with NPAT was further analyzed by means of 30 ns molecular dynamics performed in the NTP ensemble in the presence of explicit water molecules. All four objects experienced only small structural reorganization during the first 10 ns of the simulations, and the backbone structure remained stable afterward. For all proteins, only regions with a well-defined structure were analyzed: T1930 – R1982 for FLASH, T2156 – A2209 for YARP and A1412 – E1427 for NPAT.

Both complexes were further tuned by 10 successive rounds of optimization of side-chain rotamers with FoldX software (ver 5.0) [76]. Finally, the variation of free energy of the dimer formation caused by particular amino acid replacements was assessed using FoldX. For both structures, the effect of F1419A, L1420A, L1423A, and Y1425A replacements in NPAT has been analyzed in the context of three independent pulldown experiments (Appendix A). The observed exponential relation between ΔΔG and pulldown results (blue line) validates the proposed structural models of the NPAT complexes with FLASH and YARP. The only disagreement identified for L1423A replacement in FLASH–NPAT complex most probably reflects suboptimal packing of residues proximal to L1423 in the complex.

### 4.7. Modeling the Interactions of FLASH–NPAT and YARP–NPAT Complexes with DNA

The ability of FLASH and YARP alone or in a complex with NPAT to bind DNA was assessed using a combination of modeling by homology and 30 ns molecular dynamics (NPT ensemble, explicit water molecules). Initial structures of FLASH and YARP bound to DNA were obtained using three complexes of Myb2 domains with DNA as templates (3osf, 3osg, and 5lxu). For each template, up to 5 alignments with the target sequence were used, and up to 50 different conformations were tested for each modeled loop. The resulting models were evaluated according to structural quality (dihedral distribution, backbone, and side-chain packing). The top-scoring model selected from those that covered the largest part of the target sequence was used as a template to build a hybrid model, which was further interactively improved using the best fragments (e.g., loops) identified among the highly-scored single-template models. Both models were further subjected to molecular dynamics simulations. In parallel, the C-terminal NPAT fragment was placed in the FLASH/YARP–DNA complexes using structural alignments with corresponding FLASH/YARP complexes with NPAT. This approach yielded ternary structures of FLASH/YARP–NPAT–DNA complexes, which after short pre-optimization were subjected to 70 ns molecular dynamics.

### 4.8. GST Pull-Down Assay

For pull-down assays, the DNA of the C-terminal fragments of FLASH and YARP were cloned into the pET42a vector and expressed in bacteria as fusions with the N-terminal GST tag and two C-terminal HisTag${}_{6}$. The proteins were purified on Glutathione Sepharose 4B resin (GE Healthcare) according to the manufacturer’s protocol and Size Exclusion Chromatography (SEC). Binding efficiency was expressed as the percentage of the radioactive input (total amount of 35S-labeled NPAT used in the experiment) that was bound by FLASH–GST or YARP–GST (detected on glutathione beads). The percentage calculated for FLASH and YARP was reduced by subtracting the background level obtained by using GST alone and normalized as 100% for wild type NPAT. Each NPAT mutant was tested next to wild type NPAT in at least three independent experiments and the calculated data were plotted as a bar graph.

## Figures and Tables

**Figure 1 ijms-21-05268-f001:**
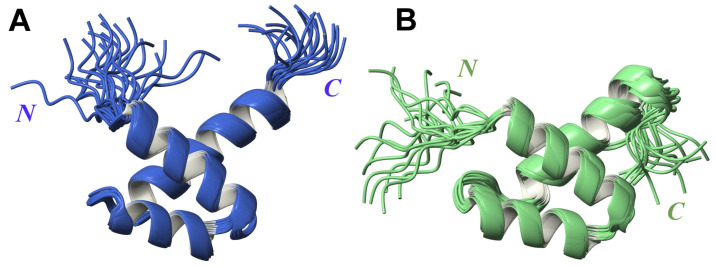
Superposition of 20 lowest-energy 3D structures obtained for FLASH (**A**) and YARP (**B**) fragments.

**Figure 2 ijms-21-05268-f002:**
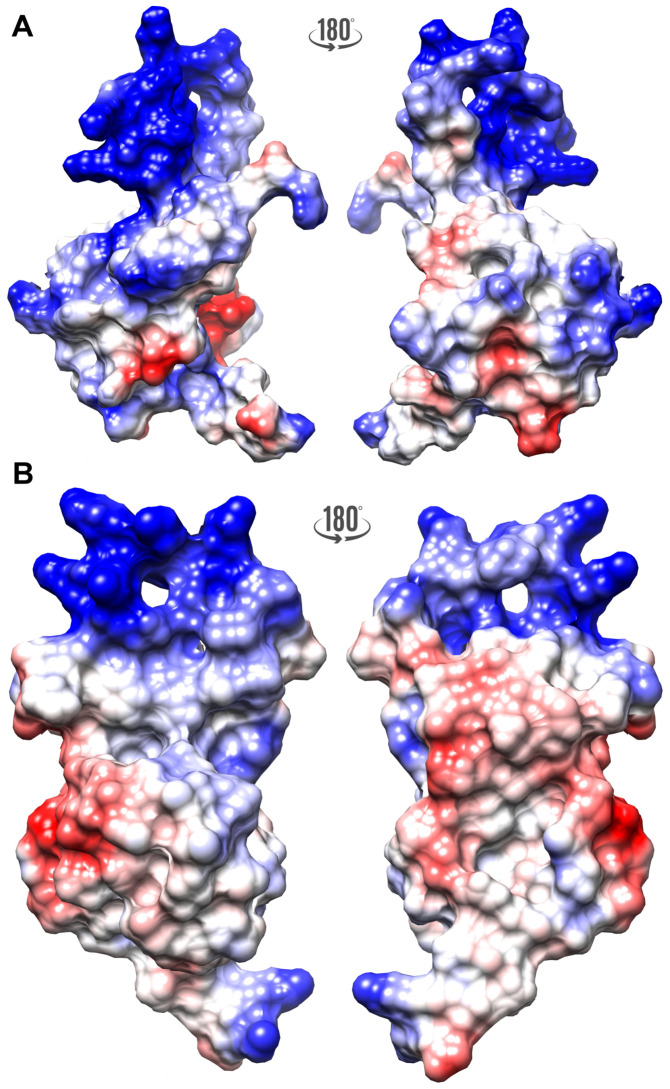
The electrostatic surface potential calculated for the FLASH (**A**) and YARP (**B**) fragments calculated at pH 6.2 with PDB2PQR [43], PROPKA [44], and APBS [45]. The negative and positive potential are highlighted as red and blue, respectively.

**Figure 3 ijms-21-05268-f003:**
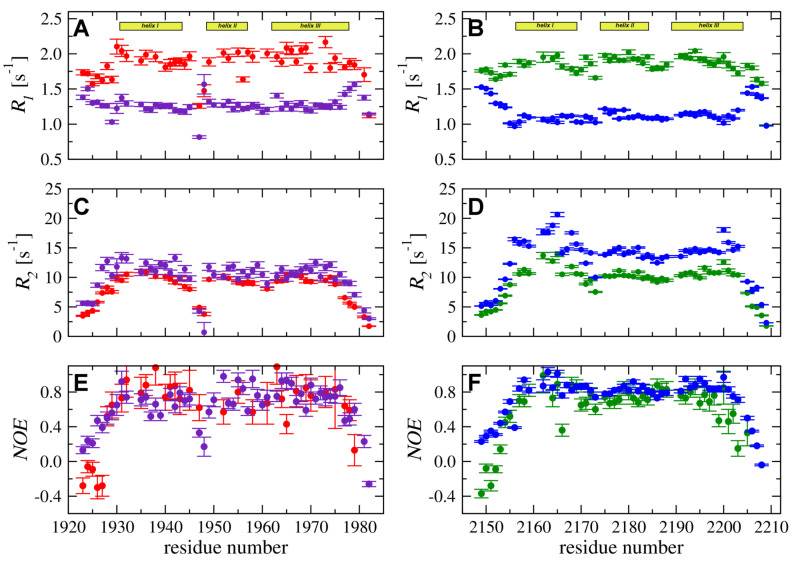
${}^{15}$N longitudinal ($R_{1}$) (**A**,**B**) and transverse ($R_{2}$) (**C**,**D**) relaxation rates and ${}^{1}$H-${}^{15}$N heteronuclear NOEs (**E**,**F**) were acquired at 298 K and magnetic field strengths of 11.7 T (red for FLASH and green for YARP) and 18.8 T (purple for FLASH and blue for YARP).

**Figure 4 ijms-21-05268-f004:**
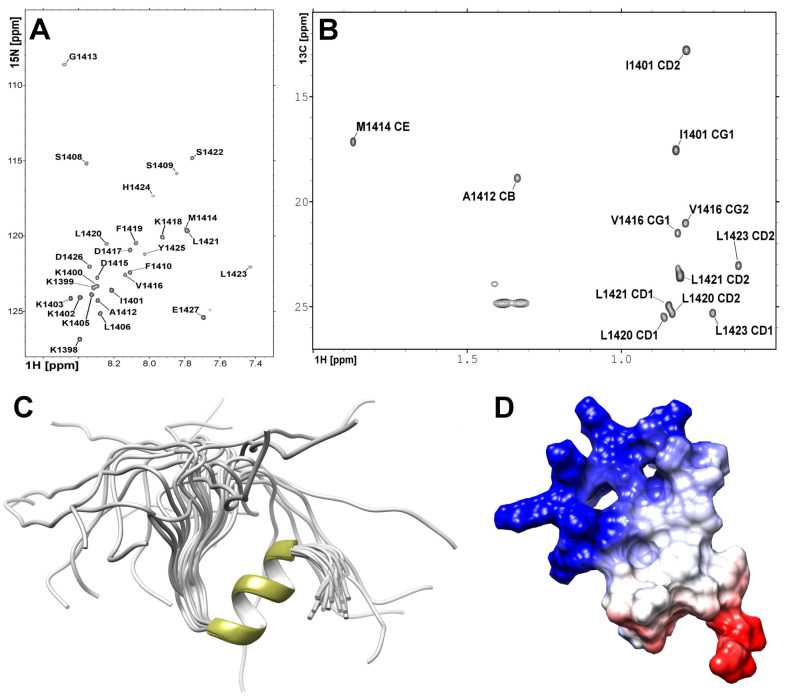
Heteronuclear NMR spectra acquired for NPAT peptide at 298 K utilizing Agilent DDR2 800 NMR spectrometer. (**A**) ${}^{1}$H-${}^{15}$N HSQC spectrum. (**B**) Region of ${}^{1}$H-${}^{13}$C HSQC spectrum representing resonances of the CH${}_{3}$ methyl groups. (**C**) The 3D high-resolution structure of the NPAT peptide in solution. (**D**) The electrostatic potential of the NPAT peptide, where negative and positive potentials are highlighted in red and blue, respectively.

**Figure 5 ijms-21-05268-f005:**
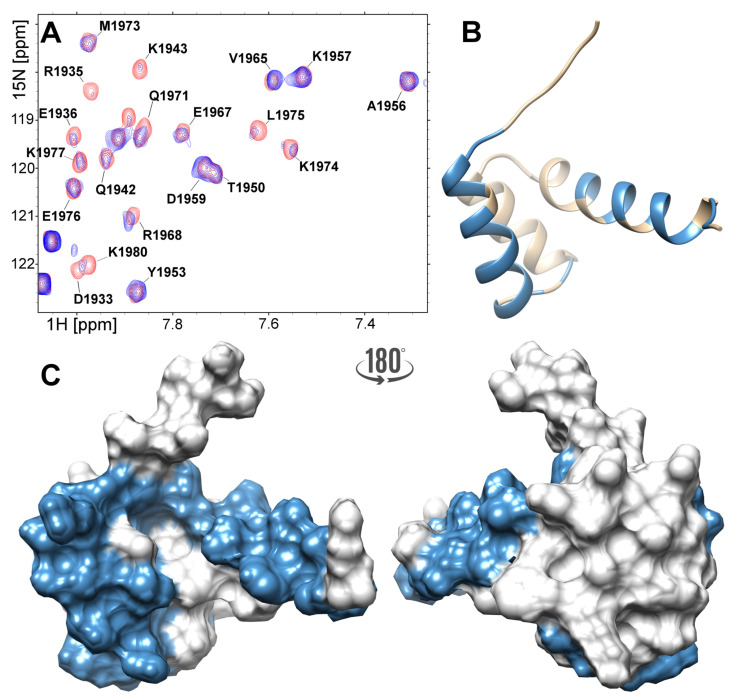
(**A**) Region of the ${}^{1}$H–${}^{15}$N HSQC spectra acquired for the 300 μM FLASH before (red) and after addition of the 150 μM NPAT peptide (blue). The 3D structure of the FLASH fragment presented as ribbon (**B**) and surface (**C**). Residues which revealed large chemical shifts or disappeared upon saturation with the NPAT peptide are highlighted in blue.

**Figure 6 ijms-21-05268-f006:**
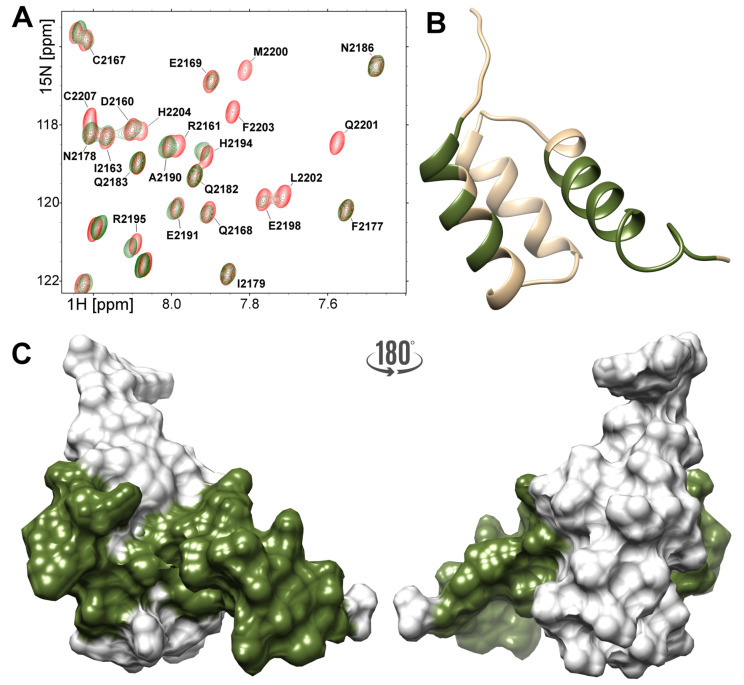
(**A**) Region of the ${}^{1}$H–${}^{15}$N HSQC spectra acquired for the 200 μM YARP (red) and after addition of the 100 μM NPAT peptide (green). A ribbon (**B**) and surface (**C**) diagram of the C-terminal fragment of YARP. Residues those peaks shifted or disappeared under titration with NPAT peptide are highlighted.

**Figure 7 ijms-21-05268-f007:**
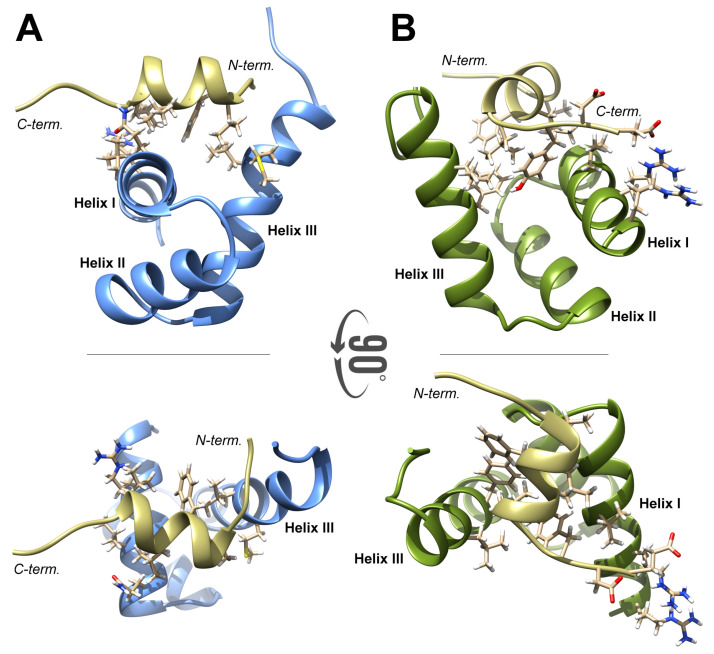
(**A**) The 3D structure of FLASH–NPAT complex. The side chains of the residues involved in complex formation (α-helix I and III in FLASH and C-terminal α-helix in NPAT) are shown. (**B**) The 3D structure of YARP–NPAT complex. Only the side chains of the residues that form binding interface are shown.

**Figure 8 ijms-21-05268-f008:**
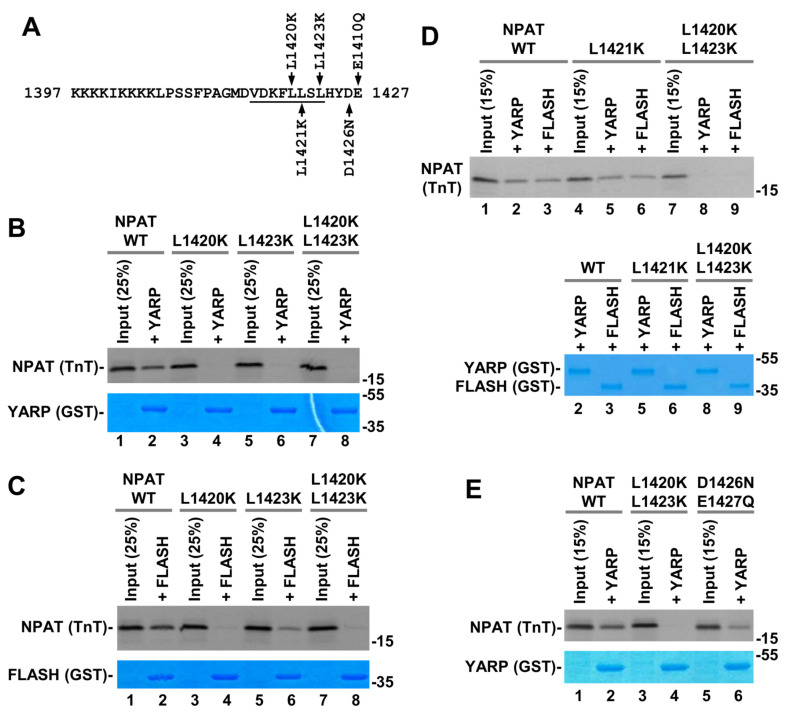
(**A**) Indicated amino acid substitutions were made in NPAT fragment encompassing the last 131 amino acids of the protein. Only the C-terminal 31 residues corresponding to the NPAT peptide used in the NMR studies are shown in the sequence. Residues that form an α-helix are underlined. (**B**–**E**) GST pull down assay. Indicated GST proteins were incubated with ${}^{35}$S-labeled wild type or mutant NPAT variants. The complexes were purified on glutathione beads, separated by SDS-polyacrylamide gel electrophoresis and GST proteins and ${}^{35}$S-labeled NPAT detected by Coomassie Blue staining or radiography, respectively.

**Figure 9 ijms-21-05268-f009:**
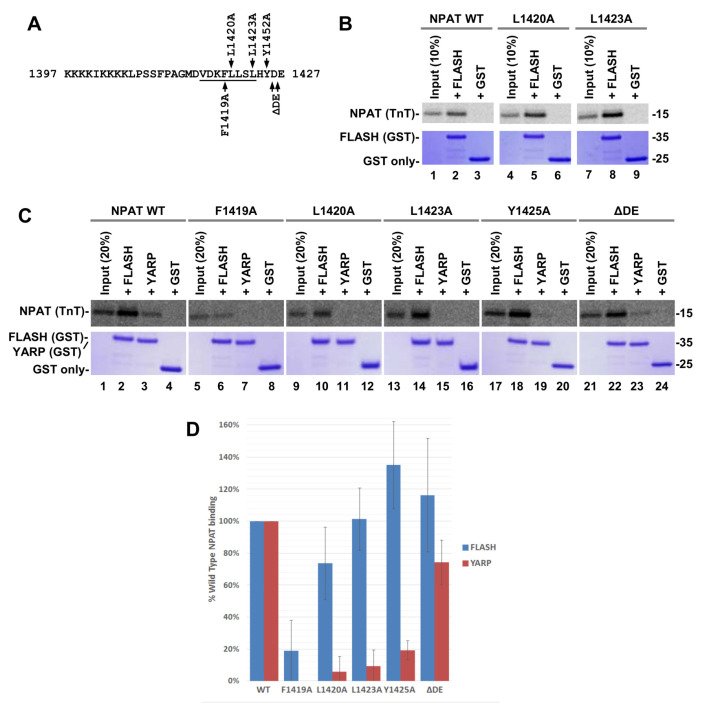
(**A**) As in Figure 8. The indicated amino acids were replaced with alanine or deleted (ΔDE). (**B**,**C**) GST pull down assay (see Figure 8). (**D**) Bar graphs calculated from three independent GST pull down experiments shown in panel **C** and panels **A,B** in Appendix A.

**Table 1 ijms-21-05268-t001:** NMR distance constraints and structural statistics for the ensemble of 20 lowest energy structures of C-terminal fragments of FLASH, YARP and NPAT.

	FLASH	YARP	NPAT
*NOESY distance constraints*			
Intraresidue	219	224	129
Sequential (|*i − j*| = 1)	147	201	106
Medium range (|*i − j*| ≤ 5)	94	141	38
Long range (|*i − j*| > 5)	81	40	9
Hydrogen bonds	188	160	10
*Torsion angles restraints* ${}^{a}$			
Backbone & side chain (ϕ / ψ / $\chi_{1}$)	129	126	50
*RMSD to main structure*			
Ordered backbone atoms ($\dot{A}$)	0.53 ± 0.14	0.66 ± 0.19	0.17 ± 0.05
Ordered heavy atoms ($\dot{A}$)	1.67 ± 0.20	1.63 ± 0.23	0.95 ± 0.26
*Ramachandran plot* ${}^{b}$			
Residues in most favored regions (%)	97.1	91.1	88.3
Residues in additionally allowed regions (%)	2.9	7.7	11.5
Residues in generously allowed regions (%)	0.0	0.1	0.2
Residues in disallowed regions (%)	0.01	0.1	0.0
*RMS Z-score* ${}^{c}$			
Bond lengths ($\dot{A}$)	0.253 ± 0.038	−0.502 ± 0.100	0.056 ± 0.082
Bond angles	0.985 ± 0.115	−0.534 ± 0.111	0.062 ± 0.105
Dihedral angles	0.589 ± 0.054	0.594 ± 0.111	−0.137 ± 0.178
Side chains planarity	1.242 ± 0.013	0.835 ± 0.075	0.267 ± 0.474
Non-bonded interactions	−0.013 ± 0.102	−0.205 ± 0.101	−0.538 ± 0.392

${}^{a}$ Evaluated with the TALOS-N software [38], ${}^{b}$ Evaluated with the PROCHECK-NMR software [40], ${}^{c}$ Calculated with the WhatIf software [41] included in YASARA [42].

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
