# Peer review of "Structural Analysis of the SANT/Myb Domain of FLASH and YARP Proteins and Their Complex with the C-Terminal Fragment of NPAT by NMR Spectroscopy and Computer Simulations"

_ijms, 2020, doi:10.3390/ijms21155268_

Round 1

Reviewer 1 Report

The authors have reported the solution structures by NMR of the structurally homologous C-terminal domains (SANT) of the two YARP and FLASH proteins reported to associate with the Nuclear Protein, Ataxia-Telangiectasia Locus (NPAT). NPAT is localized into the nucleus and is reported to bind the histone locus bodies (HLBs) controlling histone gene expression, therefore binding by these proteins has an interest for the gene expression regulation mediated by NPAT.

The authors also investigate the structure of the last 31 residues of NPAT, perform shift perturbation assays by NMR of the YARP and FLASH C-terminal domains, perform binding assays by GST pull-down and propose models of the complexes as determined by molecular docking studies. A number of mutants have also been generated and tested to support the structures of the complexes

The study is well designed, well executed and very well presented. A number of biophysical studies with relevant data are provided to substantiate the conclusions drawn.

The study deserves publication in the present form.

Author Response

The reviewer does not have any suggestions for improving the manuscript

Reviewer 2 Report

In this paper, the authors showed structural analysis of the domain of FLASH, YARP and their complex with NPAT by using NMR technique and in silico modeling. They also demonstrated that each domain of FLASH or YARP binds to NPAT, and produced two distinct complexes even though sharing a significant amino acid similarity. There are some comments and suggestions should be addressed.

  1. Page 7, line 162-164, the authors suggested “The assignment of the NPAT 1H resonances was performed based on the combination of homonuclear 2D TOCSY (mixing time 80 ms) and NOESY (mixing time 120 ms) spectra”. The authors should add the two spectra to support their experiment results.
  2. The numbers of the X-axis and Y-axis for the 2D NMR spectra should be enlarged. Also, 15N, 13C, 1H should be marked in each axis. Furthermore, the peaks assignment in Figure 4a (such as S1408) are quite small for observation. The author should change the graphic size to make the size of the words easier to recognize (like Fig 4b).
  3.  Figure 5A, Figure 6A, Figure S7 and S8, the authors should offer the ratio of NPTA to FLASH or YAPR in the Figure caption. 
  4. Some orders of Figures should be rearrangement. Such as the content of Figure 6B should precede that of Figure 6C.

Correct the following.
In page 12, line 267, “Figure 8B” should be corrected to “Figure 8D”.
In page 12, line 272, “Figure 8” should be corrected to “Figure 8A”.
In page 12, line 273, “Figure 8D” should be corrected to “Figure 8E”.

Round 2

Reviewer 2 Report

This manuscript has been improved and the work deserves publication.